# Feature Encodings for Gradient Boosting with Automunge

**Nicholas J. Teague**
Automunge
Altamonte Springs, FL 32714

## Abstract

Automunge is a tabular preprocessing library that encodes dataframes for supervised learning. When selecting a default feature encoding strategy for gradient boosted learning, one may consider metrics of training duration and achieved predictive performance associated with the feature representations. Automunge offers a default of binarization for categoric features and z-score normalization for numeric. The presented study sought to validate those defaults by way of benchmarking on a series of diverse data sets by encoding variations with tuned gradient boosted learning. We found that on average our chosen defaults were top performers both from a tuning duration and a model performance standpoint. Another key finding was that one hot encoding did not perform in a manner consistent with suitability to serve as a categoric default in comparison to categoric binarization. We present here these and further benchmarks.

## 1 Introduction

The usefulness of feature engineering for applications of deep learning has long been considered a settled question in the negative, as neural networks are on their own universal function approximators (Goodfellow et al., 2016). However, even in the context of deep learning, tabular features are often treated with some form of encoding for preprocessing. Automunge (Teague, 2021b) is a platform for encoding dataframes developed by the authors. This python library was originally built for a simple use case of basic encoding conventions for numeric and categoric features, like z-score normalization and one-hot encodings. Along the iterative development journey we began to flesh out a full library of encoding options, including a series of options for numeric and categoric features that now include scenarios for normalization, binarization, hashing, and missing data infill under automation. Although it was expected that these range of encoding options would be superfluous for deep learning, that does not rule out their utility in other paradigms which could range from simple regression, support vector machines, decisions trees, or as will be the focus of this paper, gradient boosting.

The purpose of this work is to present the results of a benchmarking study between alternate encoding strategies for numeric and categoric features for gradient boosted tabular learning. We were particularly interested in validating the library's default encoding strategies, and found that in both primary performance metrics of tuning duration time and model performance the current defaults under automation of categoric binarization and numeric z-score normalization demonstrated merit to serve as default encodings for the Automunge library. We also found that in addition to our default binarization, even a frequency sorted variant of ordinal encoding on average outperformed one hot encoding.

Has it Trained Yet? Workshop at the Conference on Neural Information Processing Systems (NeurIPS 2022).

## 2 Automunge

Automunge (Teague, 2021b) is an open source python library, available now for pip install, built on top of Pandas (McKinney, 2010), Numpy (Harris et al., 2020), SciKit-learn (Pedregosa et al., 2011), and Scipy (Virtanen et al., 2020). It takes as input tabular data received in a tidy form, meaning one column per feature and one row per sample, and returns numerically encoded sets with infill to missing points, thus providing a push-button means to feed raw tabular data directly to machine learning. The extent of derivations may be minimal, such as numeric normalizations and categoric binarizations under automation, or may include more elaborate univariate transformations, including aggregated sets thereof. Generally speaking, the transformations are performed based on a fit to properties of features in a designated training set, and then that same basis may be used to consistently and efficiently prepare subsequent test data, as may be intended for use in inference or for additional training data preparation.

The interface is channeled through two master functions, automunge(.) and postmunge(.). The automunge(.) function receives a training set and if available also a consistently formatted test set, and returns a collection of dataframes intended for training, validation, and inference — each of these aggregations further segregated into subsets of features, index, and label sets. A validation set, if designated by ratio of partitioned data from the training set, is segregated from the training data prior to transformations and then consistently prepared on the train set basis to avoid data leakage between training and validation. The function also returns a populated python dictionary, which we call the postprocess_dict, recording steps and parameters of transformations. This dictionary may then be passed along with subsequent test data to the postmunge(.) function for consistent preparations on the train set basis, as for instance may be applied sequentially to streams of data. Because it makes use of train set properties evaluated during a corresponding automunge(.) call instead of directly evaluating properties of the test data, preparing data in the postmunge(.) function can be very efficient.

There is a built in extensive library of feature encodings to choose from. Numeric features may be assigned to any range of transformations, normalizations, and bin aggregations. Sequential numeric features may be supplemented by proxies for derivatives (Teague, 2022b). Categoric features may be encoded as ordinal, one hot, binarization, hashing, or even parsed categoric encoding (Teague, 2022c) with an increased information retention in comparison to one hot encoding by a vectorization as a function of grammatical structure shared between entries. Categoric sets may be collectively aggregated into a single common binarization. Categoric labels may have label smoothing applied (Szegedy et al., 2016), or fitted smoothing where null values are fit to class distributions. Sets of transformations to be directed at targeted features can be assembled which include generations and branches of derivations by making use of our family tree primitives (Teague, 2021b), as can be used to redundantly encode a feature in multiple configurations of varying information content. Such transformation sets may be accessed from those predefined in an internal library for simple assignment or alternatively may be custom configured. Even the transformation functions themselves may be custom defined from a very simple template. Through application statistics of the features are recorded to facilitate detection of distribution drift. Inversion is available to recover the original form of data found preceding transformations, as may be used to recover the original form of labels after inference. Missing data is imputed by auto ML models trained on surrounding features (Teague, 2022a). Noise may be channeled into feature encodings for non-deterministic inference, as may include stochastic perturbations sampled from quantum circuits (Teague, 2022d).

## 3 Gradient Boosting

Gradient boosting (Friedman, 2000) refers to a paradigm of decision tree learning (Quinlan, 1986) similar to random forests (Breiman, 2001) but in which the optimization is boosted by recursively training an iteration's model objective to correct the performance of the preceding iteration's model. It is commonly implemented in practice by the XGBoost library (Chen & Guestrin, 2016) for GPU acceleration, although there are architecture variations available for different fortes, like LightGPM (Ke et al., 2017) which may train faster on CPU's than XGBoost (with a possible performance tradeoff).

Gradient boosting has traditionally been found as a winning solution for tabular modality competitions on the Kaggle platform, and its competitive efficacy has even been demonstrated for more sophisticated applications like time series sequential learning when used for window based regression

(Elsayed et al., 2021). Recent tabular benchmarking papers have found that gradient boosting may still mostly outperform sophisticated neural architectures like transformers (Gorishniy et al., 2021), although even a vanilla multi layer perceptron neural network could have capacity to outperform gradient boosting with comprehensively tuned regularizers (Kadra et al., 2021). Gradient boosting can also be expected to have higher latency inference than neural networks (Borisov et al., 2021).

Conventional wisdom is that one can expect gradient boosting models to have capacity for better performance than random forests for tabular applications but with a tradeoff of increased probability of overfitting without hyperparameter tuning (Howard & Gugger, 2020). With both more sensitivity to tuning parameters and a much higher number of parameters in play than random forest, gradient boosting usually requires more sophistication than a simple grid or random search for tuning. One compromise method available is for a sequential grid search through different subsets of parameters (Jain, 2016), although more automated and even parallelized methods are available by way of black box optimization libraries like Optuna (Akiba et al., 2019). There will likely be more improvements to come both in libraries and tuning conventions, this is an active channel of industry research.

## 4 Feature Encodings

Feature encoding refers to feature set transformations that serve to prepare the data for machine learning. Common forms of feature encoding preparations include normalizations for numeric sets and one hot encodings for categoric, although some learning libraries may accept categoric features in string representations for internal encodings. Before the advent of deep learning, it was common to supplement features with alternate representations of extracted information or to combine features in some fashion. Such practices of feature engineering are sometimes still applied in gradient boosted learning, and it was one of the purposes of these benchmarks to evaluate benefits of the practice in comparison to directly training on the data.

An important distinction of feature encodings can be considered as those that can be applied independent of an esoteric domain profile verses those that rely on external structure. An example could be the difference between supplementing a feature with bins derived based on the distribution of populated numeric values verses extracting bins based on an external database lookup. In the case of Automunge, the internal library of encodings follows almost exclusively the former, that is most encodings are based on inherent numeric or string properties and do not consider adjacent properties that could be inferred based on relevant application domains. (An exception is made for date-time formatted features which under automation automatically extract bins for weekdays, business hours, holidays, and redundantly encodes entries based on cyclic periods of different time scales (London, 2016).) The library includes a simple template for integrating custom univariate transformations (Teague, 2021a) if a user would like to integrate into a pipeline alternate conventions.

### 4.1 Numeric

Numeric normalizations [Appendix A] in practice are most commonly applied similar to our default of z-score 'nmbr' (subtract mean and divide by standard deviation) or min-max scaling 'mnmx' (converting to range between 0–1). Other variations that may be found in practice include mean scaling 'mean'(subtract mean and divide by min max delta), and max scaling 'mxab' (divide by feature set absolute max). More sophisticated conventions may convert a distribution shape in addition to the scale, such as the box-cox power law transformation 'bxcx' (Box & Cox, 1964) or Scikit-Learn's (Pedregosa et al., 2011) quantile transformer 'qttf', which both may serve the purpose of converting a feature set to closer resemble a Gaussian distribution. In general, numeric normalizations are more commonly applied for learning paradigms other than those based on decision trees, where for example in neural networks they serve the purpose of normalizing gradient updates across features. We did find that the type of normalizations applied to numeric features appeared to impact performance, and we will present these findings below.

### 4.2 Categoric

Categoric encodings [Appendix B] are most commonly derived in practice as a one hot encoding, where each unique entry in a received feature is translated to boolean integer activations in a dedicated column among a returned set thereof. The practice of one hot encoding has shortcomings in the

high cardinality case (where a categoric feature has an excessive number of unique entries), which in the context of gradient boosting may be particularly impactful as an inflated column count impairs latency performance of a training operation — or when the feature is targeted as a classification label may even cause training to exceed memory overhead constraints. The Automunge library attempts to circumvent this high cardinality edge case in two fashions, first by defaulting to a binarization encoding instead of one hot, and second by distinguishing highest cardinality sets for a hashed encoding (Moody, 1988) (Weinberger et al., 2009) (Teague, 2020) which may stochastically consolidate multiple unique entries into a shared ordinal representation for a reduced number of unique entries.

The library default of categoric binarization '1010' refers to translating each unique entry in a received feature to a unique set of zero, one, or more boolean integer activations in a returned set of boolean integer columns. Where one hot encoding may return a set of n columns for n unique entries, binarization will instead return a smaller count of log2(n) rounded up to nearest integer. We have previously seen the practice discussed in the blogging literature, such as (Ravi, 2019), although without validation as offered herein.

A third common variation on categoric representations includes ordinal encodings, which simply refers to returning a single column encoding of a feature with a distinct integer representation for each unique entry. Variations on ordinal encodings in the library may sort the integer representations by frequency of the unique entry 'ord3' or based on alphabetic sorting 'ordl'.

Another convention for categoric sets unique to the Automunge library we refer to as parsed categoric encodings 'or19' (Teague, 2022c). Parsed encodings search through tiers of string character subsets of unique entries to identify shared grammatical structure for supplementing encodings with structure derived from a training set basis. Parsed encodings are supplemented with extracted numeric portions of unique entries for additional information retention in the form received by training.

## 5    Benchmarking

The benchmarking sought to evaluate a range of numeric and categoric encoding scenarios by way of two key performance metrics, training time and model performance. Training was performed over the course of 1.5 weeks on a Lambda workstation with AMD 3970X processor, 128Gb RAM, and two Nvidia 3080 GPUs. Training was performed by way of XGBoost tuned by Optuna with 5-fold fast cross-validation (Swersky et al., 2013) and early stopping criteria of 50 tuning iterations without improvement. Performance was evaluated against a partitioned 25% validation set based on a f1 score performance metric, which we understand is a good default for balanced evaluation of bias and variance performance of classification tasks (Stevens et al., 2020). This loop was repeated and averaged across 5 iterations and then repeated and averaged across 31 tabular classification data sets sourced from the OpenML benchmarking repository (Vanschoren et al., 2014). Rephrasing for clarity, the reported metrics are averages of 5 repetitions of 31 data sets for each encoding type as applied to all numeric or categoric features for training. The distribution bands shown in the figures are across the five repetitions. The data sets were selected for diverse tabular classification applications with in-memory scale training data and tractable label cardinality.

We found that these benchmarks gave us comfort in the Automunge library's defaults of numeric z-score normalization and categoric binarization. An interesting result was the outperformance of categoric binarization in comparison to one-hot encoding, as the latter is commonly used in mainstream practice as a default. Further dialogue for the interpretation of the results presented in Figures 1 and 2 are provided in [Appendix A, B].

## 6    Conclusion

We hope that these benchmarks may have provided some level of user comfort by validating the default encodings applied under automation by the Automunge library of z-score normalization and categoric binarization, both from a training time and model performance standpoint. If you would like to try out the library we recommend the tutorials folder found on GitHub (Teague, 2021b) as a starting point.

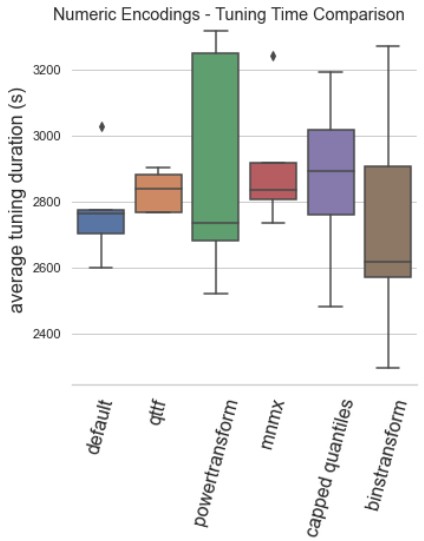
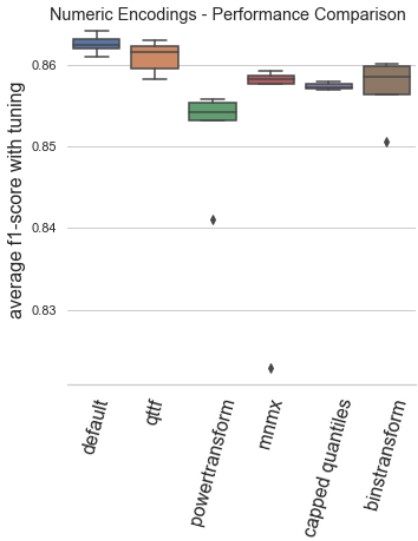

(a) Numeric tuning time comparison

(b) Numeric model performance comparison

Figure 1: Numeric Results

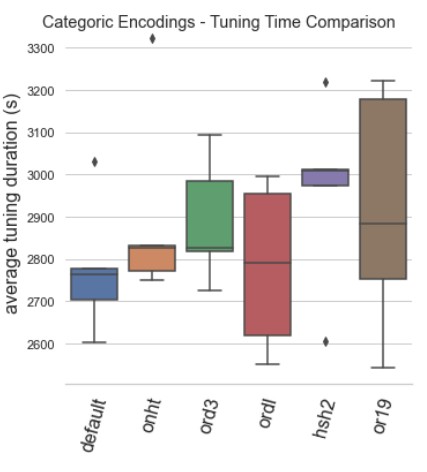
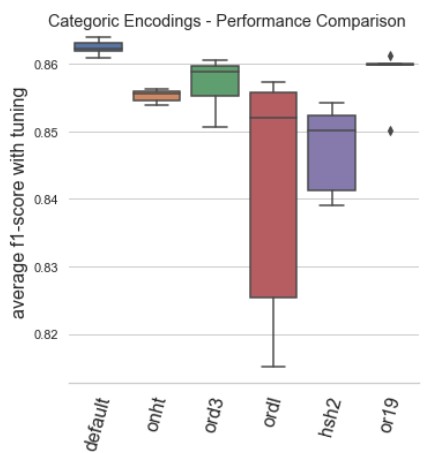

(a) Categoric tuning time comparison

(b) Categoric model performance comparison

Figure 2: Categoric Results

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

| | feature | feature_nmbr | feature_qttf | feature_nmbr | feature_mnmx | feature_mnm4 |
|---|---|---|---|---|---|---|
| 0 | 1 | -0.805313 | -0.67449 | -0.805313 | 0.076923 | 0.075472 |
| 1 | 13 | 1.495582 | 5.199337 | 1.495582 | 1 | 1 |
| 2 | 5 | -0.038348 | 0 | -0.038348 | 0.384615 | 0.389937 |
| 3 | 0 | -0.997054 | -5.199337 | -0.997054 | 0 | 0 |
| 4 | 7 | 0.345134 | 0.67449 | 0.345134 | 0.538462 | 0.54717 |

| | feature | feature_nmbr | feature_nmbr_bint_0 | feature_nmbr_bint_1 | feature_nmbr_bint_2 | feature_nmbr_bint_3 | feature_nmbr_bint_4 | feature_nmbr_bint_5 |
|---|---|---|---|---|---|---|---|---|
| 0 | 1 | -0.805313 | 0 | 0 | 1 | 0 | 0 | 0 |
| 1 | 13 | 1.495582 | 0 | 0 | 0 | 0 | 1 | 0 |
| 2 | 5 | -0.038348 | 0 | 0 | 1 | 0 | 0 | 0 |
| 3 | 0 | -0.997054 | 0 | 0 | 1 | 0 | 0 | 0 |
| 4 | 7 | 0.345134 | 0 | 0 | 0 | 1 | 0 | 0 |

Figure 3: Numeric Encodings

# A Numeric Encodings

## A.1 default

- defaults for Automunge under automation as z-score normalization ('nmbr' code in the library)
- The default encoding was validated both from a tuning duration and a model performance standpoint as top performing scenario on average.

## A.2 qttf

- Scikit-Learn QuantileTransformer with a normal output distribution
- The quantile distribution conversion did not perform as well on average as simple z-score normalization, although it remained a top performer.

## A.3 powertransform

- the Automunge option to conditionally encode between 'bxcx', 'mmmx', or 'MAD3' based on distribution properties (via library's powertransform=True setting)
- This was the worst performing encoding scenario, which at a minimum demonstrates that the heuristics and statistical measures currently applied by the library to conditionally select types of encodings could use some refinement.

## A.4 mnmx

- min max scaling 'mnmx' which shifts a feature distribution into the range 0–1
- This scenario performed considerably worse than z-score normalization, which we expect was due to cases where outlier values may have caused the predominantly populated region to get "squished together" in the encoding space.

## A.5 capped quantiles

- min max scaling with capped outliers at 0.99 and 0.01 quantiles ('mnm3' code in library)
- This scenario is best compared directly to min-max scaling, and demonstrates that defaulting to capping outliers did not benefit performance on average.

## A.6 binstransform

- z-score normalization supplemented by 5 one hot encoded standard deviation bins (via library's binstransform=True setting)
- In addition to a widened range of tuning durations, the supplemental bins did not appear to be beneficial to model performance for gradient boosting.

| | feature | feature_1010_0 | feature_1010_1 | feature_onht_0 | feature_onht_1 | feature_onht_2 | feature_ord3 | feature_ordl | feature_hsh2 |
|---|---|---|---|---|---|---|---|---|---|
| 0 | twenty one | 1 | 0 | 1 | 0 | 0 | 1 | 2 | 5 |
| 1 | twenty two | 1 | 1 | 0 | 1 | 0 | 2 | 3 | 5 |
| 2 | twenty one | 1 | 0 | 1 | 0 | 0 | 1 | 2 | 5 |
| 3 | twenty two | 1 | 1 | 0 | 1 | 0 | 2 | 3 | 5 |
| 4 | three | 0 | 1 | 0 | 0 | 1 | 3 | 1 | 4 |

| | feature | feature_or19_sp13_ord3 | feature_or19_sp13_sp10_ord3 | feature_or19_nmc8_nmbr | feature_or19_1010_0 | feature_or19_1010_1 |
|---|---|---|---|---|---|---|
| 0 | twenty one | 1 | 1 | 0 | 1 | 0 |
| 1 | twenty two | 1 | 1 | 0 | 1 | 1 |
| 2 | twenty one | 1 | 1 | 0 | 1 | 0 |
| 3 | twenty two | 1 | 1 | 0 | 1 | 1 |
| 4 | three | 2 | 1 | 0 | 0 | 1 |

Figure 4: Categoric Encodings

## B  Categoric Encodings

### B.1  default

- defaults for Automunge under automation for categoric binarization ('1010' code in the library)
- The default encoding was validated as top performing both from a tuning duration and a model performance standpoint.

### B.2  onht

- one hot encoding
- The model performance impact was surprisingly negative compared to the default considering this is often used as a default in mainstream practice. Based on this benchmark we recommend discontinuing use of one-hot encoding outside of special use cases (like e.g. for purposes of feature importance analysis).

### B.3  ord3

- ordinal encoding with integers sorted by category frequency 'ord3'
- Sorting ordinal integers by category frequency instead of alphabetic significantly benefited model performance, in most cases lifting ordinal above one hot encoding although still not in the range of the default binarization.

### B.4  ordl

- ordinal encoding with integers sorted alphabetically by category 'ordl'
- Alphabetic sorted ordinal encodings (as is the default for Scikit-Learn's OrdinalEncoder) did not perform as well, we recommend defaulting to frequency sorted integers when applying ordinal.

### B.5  hsh2

- hashed ordinal encoding (library default for high cardinality categoric 'hsh2')
- This benchmark was primarily included for reference, it was expected that as some categories may be consolidated there would be a performance impact for low cardinality sets. The benefit of hashing is for high cardinality which may otherwise impact gradient boosting memory overhead.

### B.6  or19

- multi-tier string parsing 'or19' (Teague, 2022c)
- It appears that our recent invention of multi-tier string parsing succeeded in outperforming one-hot encoding and was the second top performer, but did not perform sufficiently to recommended defaulting in comparison to vanilla binarization. We recommend reserving string parsing for cases where the application may have some extended structure associated with grammatical content, as was validated as outperforming binarization for an example in the citation.

# C   Data Sets

The Benchmarking included the following tabular data sets, shown here with their OpenML ID number. A thank you to (Vanschoren et al., 2014) for providing the data sets and (Kadra et al., 2021) for inspiring the composition.

- Click prediction / 233146
- C.C.FraudD. / 233143
- sylvine / 233135
- jasmine / 233134
- fabert / 233133
- APSFailure / 233130
- MiniBooNE / 233126
- volkert / 233124
- jannis / 233123
- numerai28.6 / 233120
- Jungle-Chess-2pcs / 233119
- segment / 233117
- car / 233116
- Australian / 233115
- higgs / 233114
- shuttle / 233113
- connect-4 / 233112
- bank-marketing / 233110
- blood-transfusion / 233109
- nomao / 233107
- ldpa / 233106
- skin-segmentation / 233104
- phoneme / 233103
- walking-activity / 233102
- adult / 233099
- kc1 / 233096
- vehicle / 233094
- credit-g / 233088
- mfeat-factors / 233093
- arrhythmia / 233092
- kr-vs-kp / 233091

