# OpenReview forum: "Feature Encodings for Gradient Boosting with Automunge"
_NeurIPS.cc/2022/Workshop/HITY — HITY Workshop NeurIPS 2022_

### Official Review · Reviewer_VSU9 · 2022-10-05
**Nice Benchmarking for the Feature Encoding in Automunge**

**Rating:** 1
**Confidence:** 3

**Review:**

This paper benchmarks different feature encoding strategies for Gradient Boosting with Automunge, and comes to the conclusion that the Default values are the best choice. The result supports the design of the Automunge package and the text is reasonably clear.

---

### Official Review · Reviewer_sf9V · 2022-10-12

**Rating:** 1
**Confidence:** 2

**Review:**

**Summary:** The paper empirically investigates the impact of different feature
encodings for tabular (numeric and categorical) data on the performance of
gradient boosted learning.

**Strengths, Weaknesses & Questions:**
- Line 1-11: In my opinion, the abstract is far too technical and detailed. A
more high-level description would help non-experts (like myself) better
understand the context of the work.
- Line 16-17: The authors explicitly say that they developed the `Automunge`
package. Since submissions are supposed to be anonymous, this is not ideal. This
issue could be circumvented by citing the Automunge paper normally (with
authors), but omitting the *developed by the authors* statement.
- Section 3.1: Some of the encoding strategies you mention and explain (e.g.
`bxcx` and `mxab`) are not part of your empirical evaluation (Figure 1). On the
other hand, some strategies like *capped quantiles* you actually used are not
explained (at least not in the main text). In my opinion, Section 3.1 should
include all strategies used in the benchmark or at least refer to the appendix.
- Line 133-137: In my opinion, this section could be improved by accompanying
the observations given here with an interpretation and possible explanations
for the results. This would strengthen the contribution of the paper.

**Minor:**
- Line 34-36: In this section, the authors describe what *gradient boosting* is.
Since I don't have any experience with this method, I would have appreciated
more detailed explanations.
- Line 41, 49 and 56: Some formulations (like *traditionally*, *conventional
wisdom*, *likely*) are quite vague and should be supported by evidence.
- Section 3.2: I think your explanations of the different encodings for
categorical data would benefit from a simple example.
- Figure 1 could be improved by using vector graphics. Also, the left and right
plot could be combined into a *performance per tuning duration* plot (with
tuning duration on the x-axis and performance on the y-axis) which would be more
compact and expressive.

---

### Decision · Program_Chairs · 2022-10-20

Accept